# Commonalities and Differences in the Experiences of Visible Minority Transnational Carer–Employees: A Qualitative Study

**DOI:** 10.3390/ijerph20186800

**Published:** 2023-09-21

**Authors:** Reemal Shahbaz, Allison Williams, Bharati Sethi, Olive Wahoush

**Affiliations:** 1Faculty of Health Sciences, McMaster University, Hamilton, ON L8S 4L8, Canada; 2Faculty of Sciences, McMaster University, Hamilton, ON L8S 4L8, Canada; awill@mcmaster.ca; 3Department of Political Studies, Trent University, Peterborough, ON K9L 0G2, Canada; bharatisethi@trentu.ca; 4School of Nursing, McMaster University, Hamilton, ON L8S 4L8, Canada; wahousho@mcmaster.ca

**Keywords:** transnational carer–employees, immigration, visible minority, carer-friendly workplace policies

## Abstract

This qualitative study explored the commonalities and differences among the experiences of visible minority Transnational Carer–Employees (TCEs) before and after COVID-19. TCEs are immigrants who live and work in the country of settlement while providing caregiving across international borders. Purposive and snowball sampling resulted in the participation of 29 TCEs of Pakistani, Syrian, African, and South American origin living in London, Ontario. Thematic analysis of the dataset using the ATLAS.ti software, Version 23.2.1., generated three themes: (1) feelings associated with transnational care; (2) employment experiences of TCEs; and (3) coping strategies for well-being. The results of the secondary analysis conducted herein suggested that there are more similarities than differences across the four cohorts. Many participants felt a sense of satisfaction at being able to fulfill their care obligations; however, a different outlook was observed among some Syrian and African origin respondents, who disclosed that managing care and work is overwhelming. Most TCEs also reported facing limited job options because of language barriers. While various interviewees experienced a lack of paid work and reduced income after COVID-19, a distinct perspective was noted from African descent TCEs as they expressed facing increased work demands after the pandemic. Participants additionally revealed four common coping strategies such as keeping busy, praying, family support, and staying active. Study implications include the promotion of Carer-Friendly Workplace Policies (CFWPs) that can facilitate the welfare of unpaid caregivers. This research is important as it may inform policymakers to create opportunities that may not only foster economic stability of TCEs and the Canadian economy, but also contribute towards a more equitable society.

## 1. Introduction

In Canada, there are approximately six million carer–employees (CEs), representing 35% of the total workforce [1]. CEs are defined as employed family members and other significant people who engage in care for individuals such as parents, siblings, spouses, and/or friends living with a physical, mental, or cognitive condition [2]. Approximately 60% of CEs in Canada manage their unpaid caregiving duties along with their employment and other family commitments [1]. Earlier research shows that there is a lack of flexibility, employer recognition for CEs, and carer-friendly workplace policies (CFWPs) that can best accommodate caregivers in the labor force [2,3,4]. Caregiving responsibilities tend to be unpredictable and complex, making them a challenge to juggle with paid work [2]. Specifically, 10% of Canadian CEs turn down employment opportunities, 26% take absences from work, and 19% experience physical and emotional health challenges [4,5]. Correspondingly, in 2018, approximately 6% of CEs left or intended to leave the workplace altogether due to caregiving duties [6]. Employed caregivers endure additional economic hardships because when they decrease their work hours due to unpaid care work, they lose some or all employee benefits like life insurance, prescription plans, dental, and other extended health benefits [1].

Moreover, the rapidly changing age structure of Canada is increasingly causing family and friends to take on caregiving roles [7]. Increased life expectancies and the transition of baby boomers into their elderly years mean that more Canadians will need assistance and services that will help them live comfortably in their senior years [5]. The 2018 Canadian General Social Survey revealed that about one in four individuals living in Canada aged 15 or above, or about 7.8 million Canadians, have provided care to at least one family member or friend who is experiencing a disability, chronic health illness, or aging needs [1]. Statistics Canada highlights that between 2016 and 2021, there was a 12% rise in the number of individuals aged 85 and above [8]. This currently represents 2.3% of the Canadian population; however, over the next two decades, it is estimated that the population aged 85 and older will triple to 2.5 million, accounting for 6.9% of the total population in Canada [8]. As life expectancy is moving beyond the age of 85, there will be a higher number of people facing long-term health conditions [8]. Consequently, the Canadian society will see a boost of CEs in the workforce, and a continued upsurge in immigration to drive population growth as well as support the Canadian economy [3,4,7]; these trends will see a growth in the number of Transnational Carer–Employees (TCEs). TCEs are immigrants who live and work in the country of relocation while providing emotional, physical, or financial care to their family members or friends across borders. For instance, this can include but is not limited to sending money and medication for loved ones in their birth country and listening to their concerns or giving them advice through calls. Many TCEs also tend to have family and other dependents that they care for in the country they immigrate to [4,7].

Currently, there are 500,000 new immigrants entering Canada annually [9]. This is one of the highest immigration rates for any nation globally [9]. Approximately eight million, or 20%, of the total population of Canada consisted of immigrants in 2022 [9]. Statistics Canada reports that 19% of Canadians identify themselves as a visible minority member. Of this, 30.9% were born in Canada whereas, 65.1% were immigrants [10]. Many immigrants are involved in transnational caregiving due to increasing mobility and the aging population worldwide [6]. In addition, it is important to note that about 61% of employed caregivers across Canada are between the ages of 45 to 64, constituting workers in their peak earning years with valuable skills and experience in their fields [5]. Thus, to improve worker retention, there is a need for employers to enhance diversity in the workplace and to create more flexibility to the challenges that unpaid caregivers face related to care and work [5]. To do this, it is essential for workplaces to implement gender sensitive and CFWPs such as the CSA B701-17 Carer-Inclusive and Accommodating Organizations Standard (The Standard) [11]. Developed in 2017 via a partnership between McMaster University and the Canadian Standards Association (CSA), The Standard provides guidelines for employers, human resource professionals, and other establishments to implement organizational changes in workplace practices to decrease work–family hardships and provide adequate support to CEs outside the workforce [11,12]. Existing scholarly work reports that having carer-inclusive and accommodating policies in places of employment can generate a positive impact for both employees and employers [1,4,13,14,15]. Outcomes include decreased stress, enhanced employee retention, lower sick leave costs, higher productivity, improved service delivery, as well as increased employee morale and commitment to the workplace [1,4,13,14,15].

This study is part of a larger research program investigating carer-friendly workplaces that examines the experiences of Indigenous, European, and Visible Minority TCEs based in London, Ontario [12]. The goal of the research program is also to scale up The Standard and develop its associated tools to be culturally competent, inclusive, and accommodating [12]. The current study is one of the 17 projects from Stream B: Healthy, Productive Work Partnership Grant, of the broader research program [12]. A research article on the same data as the present study has been published by Rottenberg [16] in the Journal of *Wellbeing, Space & Society*. The secondary analysis conducted herein generated two publishable manuscripts, the first paper (currently under review) centralized on the experiences of transnational caregiving among visible minority TCEs. The current paper further goes on to explore the commonalities and differences among visible minority TCEs.

A systematic review by Sethi et al. [17] disclosed that due to the lack of CFWPs in the workforce along with geographical separation from their families, TCEs experience negative mental health impacts such as tension, guilt, shame, anxiety, and depression. Comparatively, a study by Amin and Ingman [2] assessed the emotional impact of providing transnational caregiving to elderly parents among TCEs living in the United States. The results of the study illustrate that TCEs face role overload, psychological distress, and relational deprivation [2]. The authors specifically publicized that TCEs may experience poor physical and mental health outcomes, such as chronic anxiety and depression that can, over time, lead to severe health problems [2]. Similarly, a paper by Ilagan et al. [18] discussed that TCEs face physical and emotional health challenges because of balancing work and unpaid care across transnational boundaries. Ilgan et al. [18] used photovoice methods to highlight that TCEs may feel stressed out and constantly worry about sending money to their care-recipients.

Furthermore, a survey conducted in 2020 by the Carer Well-Being Index looked at the impact of COVID-19 on 479 caregivers employed in the Canadian workforce [1]. The findings of the survey reported that 53% of the participants in Canada did not obtain carer-friendly support from their employers and 34% faced an increase in difficulty juggling their paid work and informal caregiving responsibilities after COVID-19 [1]. Despite past research categorically reporting adverse impacts on the physical, economic, and psychological well-being of TCEs, there is still a gap in the literature on the experiences of visible minority TCEs before and after the pandemic [3,4,7]. To our knowledge, this study is the first to examine the commonalities among, and differences between, the experiences of TCEs of South American, African, Pakistani, and Syrian descent living in London, Ontario, before and after COVID-19. Moreover, in the context of this research, it is important to note that the difference between CEs and TCEs is that CEs provide care domestically in Canada whereas TCEs are immigrants that live and work in Canada but engage in caregiving outside Canadian borders. Although this study highlights the experiences of TCEs, it is crucial to recognize that CFWPs implemented in the workplace may help both CEs and TCEs.

To better understand the similarities and differences among the visible minority TCEs in this study, it is useful to grasp some of their cultural characteristics such as family size, belief systems, family cohesion, and care expectations. For starters, the average household size among Pakistani, Syrian, African, and South American families tends to be 6.4, 5, 6.9, and 4.6 persons per household, respectively [19,20,21,22]. Further, most of the individuals from these backgrounds practice Abrahamic religions. For instance, approximately 95% of the Pakistani and 90% of the Syrian population are Muslims [23,24]. In 2020, 60% of the African population was Christian, followed by 30% of Muslims [25]. Similarly, Roman Catholicism, a Christian denomination, is the largest religious group in South America and is practiced by more than 50% of the population [26]. Islam follows behind and is the second biggest religion in South America and is prevalent among minorities in most of the nations in the continent [26]. Notably, the religious and cultural belief systems of these four cultural cohorts emphasize the importance of caring for and helping those in need, especially family members [27,28,29,30]. Even though the rise in globalization has heightened the influence of Western culture, many individuals of Pakistani, Syrian, African, and South American origin believe in the institution of marriage and that it is an essential and basic unit of social life [28,30,31,32]. They hold values such as having connections and sharing a life with immediate and extended family members. Individuals from these cultures recognize the need to care for children and parents in old age [28,30,31,32]. People from Pakistani, Syrian, African, and South American cultures feel that it is their moral responsibility and obligation to provide care for their loved ones by caring for them emotionally and financially [28,30,31,32]. Even though there are more notable similarities among the four cohorts, the current research may additionally inform a culturally competent version of the noted CSA Carer-Inclusive and Accommodating Organizations Standard, as well as gender-sensitive CFWPs in workplaces in order that the welfare of TCEs and the Canadian economy may be sustained.

## 2. Materials and Methods

### 2.1. Reflexivity Statement

I, Reemal Shahbaz, conducted the secondary analysis in this study and I identify as an able-bodied, cis-gendered female, with a Pakistani origin. I immigrated to Canada with my parents and siblings when I was nine years old in March of 2010. I am currently 22 years old, and my worldviews are consistent with constructivism as I believe that multiple realities can exist and that knowledge is subjective and a social construct. My undergraduate degree in Honours Life Sciences and the coursework I have completed for my graduate studies in Global Health grant me a comprehensive foundation to analyze the experiences of visible minority TCEs. My academics have trained me to emphasize cultural sensitivity, and as a result, I may be better able to recognize how diverse norms and practices impact unpaid caregiving dynamics among TCEs. I was not involved in the data collection of this research; nonetheless, while I was analyzing the data, I quickly realized that my own situations and motives align with the recruited TCEs. Like the respondents, I am a visible minority immigrant who lives and is employed in Canada whilst providing transnational caregiving to individuals in the country of my origin. I have grown up to believe that it is my religious obligation and a cultural expectation to care for my parents, siblings, extended family members, and those who are in need. Due to my educational background and personal experiences of being a TCE, I may be biased to focus more on the pressure and stress associated with caregiving and the need for caregiver-friendly and gender-sensitive policies to be implemented in Canadian workplaces. To limit my biases from impacting the research, I kept an open-minded approach and was prepared to challenge my assumptions and preconceived notions throughout the data analysis. During the write-up of the research, I kept a reflexive journal and was critically reflecting on my own thoughts and ensuring that they are not imposed on the results. Therefore, I made a deliberate effort to ensure that the voices and narratives of the study participants drive the analysis of the multifaceted experiences of employed transnational carers.

### 2.2. Ethical Approval

This study was approved by the McMaster University Research Ethics Board (MREB # 4881, 24 July 2018) and King’s University College Research Ethics Review Committee (21 June 2019). The Qualtrics software (Qualtrics, Provo, UT, USA) was used to acquire informed consent from all respondents of the research. All respondents were given an honorarium, valued at CAD 35, for their time. The participants were informed that their participation in the study would be voluntary and that they could withdraw at any time if they wished to do so. To protect anonymity, the real names of all interviewees will be kept confidential, and they are referred to in this paper by pseudonyms.

### 2.3. Study Setting

London was selected as the location of interest as it is a city that has experienced a rapid increase in immigration in the last few years [33]. In 2021, there were 103,300 individuals, or 24.8% of the total population of London, who were born in a foreign country [33]. From this, approximately 16% of the immigrants in the city identified as a visible minority. Further, the number of foreign-born people gaining permanent resident status in London has increased by more than 70% according to the latest census [34]. The top places of birth of recent immigrants in London are India, Syria, and China, accounting for 19.4%, 17.7%, and 5.0% of the newcomer immigrant population, respectively [33].

### 2.4. Approach

The philosophical orientation that informs this research is constructivism. The constructivist paradigm explains that reality is a social construct and is perceived to be subjective [35,36]. Constructivism assumes that there is a need for researchers to understand the intricate tacit experiences from the perspectives of those who live it [35]. This philosophical orientation emphasizes that knowledge is created through human experiences and thus, multiple realities can exist [35]. Constructivist research centralizes on a single phenomenon and brings forth the personal values of the participants into the research [36]. The constructivist paradigm suits this study as TCEs discuss and reveal their subjective experiences of providing care on a transnational scale, whilst being employed. In addition, the theoretical framework that underpins this research is the intersectionality theory by Kimberlé Crenshaw. This theory describes how overlapping social identities of individuals are interlinked with structural racism and oppression [37]. It is crucial to understand that oppression causes the unjust treatment or subjugation of people through the power of systems and societal norms [37]. Intersectionality explains that social identities work on many distinct levels and lead to unique experiences, challenges, and opportunities for each person [37]. Hence, to understand oppression, an understanding of how people are disadvantaged due to their multiple and overlapping social identities (such as race, class, ethnicity, age, religion, disability, sexual orientation, and more) and systems of oppression (e.g., systematic racism) is necessary [38]. Recognizing intersectionality is pivotal to battling the deeply embedded and interwoven prejudices that individuals experience in their lives [38]. The intersectionality theoretical framework guided this study as it aided in the exploration of how various socially intersecting categories such as race, ethnicity, and religion are linked in the commonalities and differences among the experiences of visible minority TCEs before and after the pandemic in London. The intersectional lens may help shed light on the complexities and experiences of visible minority TCEs such that findings may inform greater facilitation and implementation of comprehensive solutions and policies [38].

### 2.5. Study Design

A community-based and ethnographic design was used for this research to obtain knowledge of the lived experiences of visible minority TCEs residing in London, Ontario. This epistemological domain allows researchers and members of the community to work together in the research process [39]. Ethnographic methods include observations and interviews to obtain a detailed insight into the participants’ views and actions as well as how they interact with and function in their real-life environments [40]. This ethnographic research used interview methods and not observations to unravel the experiences of visible minority TCEs because interviews may allow researchers to better understand and gather in-depth information about individual experiences, traditions, and mindsets [39,40]. Moreover, community-based and ethnographic research focuses on community engagement, rapport building, collective perspectives, and cultural understanding of the members of the community [39,40].

### 2.6. Participants

To be recruited for this research, participants had to be aged 18 years or above, living in London, identifying as a visible minority, providing care to a family and/or friend living across borders, and volunteering or being at a paid employment. The inclusion criteria did not focus on whether the immigrants held permanent residency status or how long they had lived in Canada. The inclusion criteria also did not focus on whether TCEs were specifically caregiving for individuals with mental or physical illness. Participants were asked for a one-time, one-on-one interview.

### 2.7. Procedure

Research posters were placed in community centers, grocery stores, local libraries, and mosques. Recruitment posters were also sent to organizations that provide services and aid to immigrants. Data were collected from October 2019 to March 2021. Purposive and snowball sampling elicited 29 visible minority TCE participants for this study. Purposive sampling technique allows researchers to intentionally select respondents that meet the unique characteristics and experiences that are relevant to the research question [41]. Correspondingly, snowball sampling, another non-probability sampling strategy, involves identifying some initial participants that meet the study inclusion criteria and then asking them to refer other individuals who fit the eligibility criteria [41]. Through this, referrals are added, and a chain-like structure is created within the sampling [41]. The interview guide of this research consisted of questions that inquired about the experience of the TCEs in managing care and work, awareness of the CFWPs, whether COVID-19 impacted their transnational care, and how they cope with their paid and unpaid responsibilities. For example, participants were asked questions like “How did caregiving evolve for you?” and “What other responsibilities do you have while working and caregiving?”. All interviews were carried out by the research assistants (RAs) at a time and location that was mutually agreed upon. The length of interviews was approximately 60 to 90 min long, and they took place in-person prior to the pandemic and over the Zoom platform post pandemic because of COVID-19 policies. Interviews were conducted in three different languages—English, Spanish, and Arabic—and they were recorded with the consent of the participants. Spanish and Arabic interviews were translated into English and then back to the original language to ensure that they were as precise as possible. All transcripts were verified with the recording by the forward and back translation.

### 2.8. Data Analysis

The secondary analysis of the data in this study is consistent with Braun and Clarke’s [42] thematic analysis. This rigorous qualitative analysis is used to organize participant concepts into themes that capture the significant narratives on the research question. Braun and Clarke [42] describe that thematic analysis can aid researchers in interpreting and explaining patterns in the dataset in a detailed manner. For this study, an inductive or bottom-up thematic analysis was conducted using the ATLAS.ti (ATLAS.ti, Berlin, Germany) qualitative coding software. An inductive analysis means that the data in this research were coded without trying to categorize it within a pre-existing coding framework or having any preconceptions about the research question [43]. Henceforth, the themes that emerged in this study were from the dataset itself and not through paying attention to the themes that are incorporated in previous research studies [43]. As stated above, the intersectionality theory informs the analysis as it facilitates assessing how the experiences of visible minority TCEs are affected by intersecting identity markers such as religion, immigration, and ethnicity.

The thematic analysis for this study was carried out in six steps as described by Braun and Clarke [42]. The first phase comprised of the primary author getting familiarized with the data. To do this, all the data were first transferred to the ATLAS.ti coding software for analysis. Then, all transcripts and field notes were carefully read. Additionally, all the interview recordings that were in English were watched and further notes were taken. At the end of this phase, the primary author was fully immersed and informed about the depth and content of the dataset. The second phase included generating initial codes. In this step, the transcripts were re-read, and the inductive codes were applied to excerpts. Similar codes were applied to the excerpts that were representative of the same idea. In phase three, as suggested by Braun and Clarke [42], the primary author searched for themes. The goal of this stage was to identify the overarching patterns and relationships throughout the whole dataset. Similar codes were brought under one code group and then organized into a theme that revealed meaningful information about the research question. Phase four of the thematic analysis encompassed reviewing the themes. At this point, themes were checked for internal homogeneity or ensuring that they were coherent and consistent [43]. Themes were also examined for external heterogeneity and thus, it was assured that there is a distinction between one theme and another [43]. After it was established that the themes were coherent, making sense, supported in the data, and not repetitive, the first author moved on to phase five. This step incorporated defining themes. This is the last time that the themes were refined. As per the suggestions of Barun and Clarke [42], the essence of what each theme is about was defined. To do this, the primary author went back to the themes to ensure that they were succinctly named according to what aspect of the data each theme was representing. Finally, a write-up of the findings was formulated in the sixth and final phase of the analysis. To enhance the trustworthiness and credibility of the data analysis, the primary author kept an audit trail of the reflexive journals and the developed themes in the ATLAS.ti software.

## 3. Results

The data for this study consisted of information from 29 visible minority TCEs living in London, Ontario. A visible minority individual is described as a person who is not aboriginal and is non-white in colour or non-Caucasian in race [44]. One interview occurred prior to COVID-19, 15 took place post COVID-19, and 13 were conducted both before and after the pandemic. The data revealed that 14 interviewees identified as females and another 15 as males (refer to Table 1). Respondents of this study were engaged in transnational care for their immediate family (parents, children, in-laws, spouse) and/or extended family (cousins, nephews, nieces, spouse’s family) in their home nation. Participants had origins from Venezuela; Barbados; Columbia; Haiti (grouped as South America), Kenya; Uganda; Zimbabwe; Nigeria (grouped as Africa), Pakistan, and Syria (refer to Table 2). The length of residence in Canada of the TCEs varied from approximately 6 months to more than 35 years. Data brought to light that participants were involved in three types of caregiving: financial, emotional, and physical. Financial caregiving consisted of sending money through service providers such as Western Union. Emotional caregiving involved communicating via phone calls or video calls through Skype, Zoom, or WhatsApp to provide verbal support, encouragement, and advice to the care-recipients. Physical caregiving involved sending medications for diabetes, vitamins, clothes, and electronic gadgets to their country of origin for the individuals they were caring for. The results of this research illustrate that there is a lack of CFWPs in the workplace and that TCEs were not aware of them and doubted that they would ever receive support for their unpaid caregiving duties from employers. The findings of this research also revealed that there are more similarities than differences across the Pakistani, Syrian, African, and South American origin TCEs. It is possible that the differences among these four cohorts may have been concealed due to the common struggles of making ends meet and providing for immediate and extended families post pandemic. Thematic analysis of the data from all 29 participants generated three main themes: (1) feelings associated with transnational care, (2) employment experiences of TCEs, and (3) coping strategies for well-being.

### 3.1. Feelings Associated with Transnational Care

The results of this research determine that TCEs experience a wide range of emotions when providing care on a transnational scale. Prior to the pandemic, a commonality among many of the visible minority TCEs recruited for this study was that they reported feeling content, energized, and gratified about having the opportunity to give back to their family. Sabrina, a Pakistani origin TCE said: “It [caregiving] gives me peace”. On a parallel note, Adam, a Syrian descent TCE, stated: “It is a reward for sure […] you help feel happy you help them [care-recipients]”. Nogi, a TCE of African descent voiced: “I tell people this is the source of my energy, when I see that I’m able to help somebody. It’s like a force, it keeps me going”. Comparatively, even after the pandemic, respondents reported feeling a sense of tranquility, comfort, and satisfaction for being able to fulfill their obligation to provide care for their loved ones and community. For example, Helena, a TCE from a South American background mentioned: “Sending them [care-recipients] money makes me feel satisfied, so satisfied, I mean, I love, and I adore my family”. Likewise, Martina, another respondent with South American roots explained: “I feel very happy, and I feel very calm that I can provide support for my mother”. Jemmy, a TCE of African descent supports these feelings and disclosed: “I feel very, very good that my support helps them in a way that improves their lives”. Therefore, these quotes exhibit that most of the visible minority TCEs in this study experienced positive feelings about the care that they were providing to their care-recipients both before and after COVID-19.

Another commonality among the participants was that they felt the urge to provide more help and that if they were unable to, they would then experience guilt, regret, and remorse. In the interviews that took place prior to the pandemic, respondents from all four visible minority groups voiced that their financial constraints limit the extent of how much they are able to provide for their loved ones. Interviewees revealed that although they are only able to provide limited transnational care for the moment, their goal is to do more in the future. Rasha, a TCE of Syrian descent said: “I regret that I am not able to help and support them [care-recipients] and can barely send a small token of financial help”. On a similar note, Nogi mentioned: “I would love to do more […] I wish I could, you know after this school, maybe more”. Nina, a TCE with African heritage also described: “In the end maybe I might not be able to provide as much as I want for my parents and my family back in Nigeria at the moment, but after a short while I can do way more”. Comparably, the interviews that were conducted post-COVID-19 depicted results on the same wavelength, as many respondents continued to feel that the extent to which they can help was hindered by their other responsibilities. For example, Alajide, a TCE of African ethnicity, said: “As much as I would love to, I’m only limited and even my family I can’t give them the best I wish I could provide right now”. Equivalently, Anela, a TCE of South American descent declared: “I always feel like I should be doing more than what I’m doing […] I could be helping more, even if I have this job, I’m only working on the weekend and then I still can’t give enough”. Sherla, another South American TCE living in London, further established: “I wish I could do more, but due to circumstances there’s only so much you can because you have other things to take care of”. Therefore, financial hardships and other commitments are evidently impediments that mitigate the capacity of TCEs to engage in the desired amount of caregiving, and thus result in a constant desire to want to do more.

Contrarily, there were differences observed among African and Syrian descent TCEs relative to the other two cohorts regarding feelings associated with transnational care. Prior to COVID-19, some participants from African and Syrian origin revealed that they felt overwhelmed with the amount of their care duties. Nina stated: “It’s overwhelming. I’m talking about this care […] I just even need a day off”. Likewise, Alajide explained: “Aside the financial burden, it [caregiving] is emotionally draining […] I mean I have to balance the fact that I need to provide for my family as you put it, in a new country, which is more expensive”. Rasha also reinforced these notions and distinguished: “I feel I am suffocating— I feel I should be able to know when they need it [financial aid] before they have to ask [...] it really upsets me”. Correspondingly, in the interviews after COVID-19, a few TCEs from African and Syrian backgrounds continued to describe that caregiving can be frustrating. For instance, Arif, a TCE of African descent said: “Emotionally [caregiving] it’s really heavy”. Adam also stated: “It [caregiving] is just overwhelming”. Thus, while the majority of TCEs in this study explained that they are content about providing caregiving services, there was a difference of opinion among a few Syrian and African origin participants who depicted that long-distance care leads to feeling overwhelmed, emotionally drained, and frustrated.

### 3.2. Employment Experiences of TCEs

The results disclose that there are similarities and differences in the employment experiences of visible minority TCEs. In the interviews that took place before the pandemic, many participants in this study from all four ethnicities mentioned that at some point, they did jobs that did not utilize their skills and could even be categorized as demeaning in their culture. However, they continued to do it such that they could earn a living, which could help support their care-recipients. For instance, Adam stated: “If you have a job, you have to suck up something to stay in the job, because you have to take care of people”. Ammar, a TCE of Syrian roots also stated: “You just need some income, so you accept anything you get offered”. Mohammed, another participant with a Syrian background supported this concept and added: “Back home we had good jobs, respectful jobs. Here we come and we work demeaning jobs that no one else wants to work because we do not know language”. Moreover, analogous patterns were seen amongst participants even after COVID-19. Helena mentioned: “I work in any job I get […] because of the language barrier I am not able to get better jobs”. Sebastian, a South American origin TCE also explained: “I am a custodian at [redacted] […] I was a lawyer in Colombia […] finding a job wasn’t very hard. The hard part was accepting it psychologically”. Therefore, many TCEs of visible minorities residing in London unveiled that language barriers, along with the immediate need to make money for caregiving, cause them to accept employment that is not completely desirable to them.

Furthermore, COVID-19 had a significant impact on the employment experiences of Pakistani, Syrian, African, and South American descent TCEs. Many interviewees reported they would have taken on more work shifts and/or another job to ease the financial burden linked with providing international care. However, the pandemic resulted in fewer work opportunities, which hindered their ability to sustain their income. For instance, Jemmy declared: “If it wasn’t for COVID I would take another job”. Anna, a TCE from a South American background, stated: “Due to the pandemic, work has slowed down and that has reduced our income, so we’ve had to reduce the financial support that we give to this family member”. Likewise, L.A, an African descent interviewee, communicated: “I mean the COVID situation has impacted not just of course the entire world, but me personally […] they [workplace] had to let me go and um, that definitely impacted on my income”. Henceforth, COVID-19 affected the employment experience of TCEs in this study as they reported that there was less work which ultimately reduced their income and added a barrier to their financial caregiving duties.

In contrast, there were a few TCEs of African origin that had diverging experiences, as they shed light on an increase in work demand after COVID-19. Nina declared: “I would say I’m working more. Although things started off slowly, but once I got into it, there’s just a lot to do”. Correspondingly, Lulu, another participant from an African background, reported: “I was more worried about not working enough, but then I ended up working more hours because I work in a shelter so we’re more like essential service”. Similarly, Cofi, another TCE of African ethnicity, who was working as a teacher, explained that he began working more post COVID. Specifically, Cofi stated: “Preparing for classes, it increased, I made more time for that”. Hence, even though most of the TCEs of visible minority revealed that COVID-19 decreased their work and income, there were a few respondents of African descent who disclosed that they faced a higher employment load after the pandemic due to the type of work they were doing.

### 3.3. Coping Strategies for Well-Being

The results of this study acknowledge that there are four common coping strategies that Pakistani, Syrian, African, and South American origin TCEs residing in London engaged in to maintain their well-being before and after the pandemic. Visible minority TCEs across all four cohorts expressed that they tried to keep themselves busy and distracted to alleviate negative mental health symptoms. In an interview that occurred prior to the pandemic, Mohamed stated: “When I work, I forget all my burdens. It is when I do not work, that is when my mind explodes as I think about those close to me and those far away. (laughs) I get frustrated”. Similar patterns were also observed in interviews that occurred post pandemic. Luis, a TCE with South American ethnicity, said: “The ideal is to keep your mind occupied, so you don’t have bad thoughts, or get depressed or to get emotionally affected”. Sabrina also supported this idea and revealed: “I tried to keep myself busy […] to distract myself from all worries”. These statements reinforce the notion that being occupied has remained a coping mechanism for TCEs of visible minorities both before and after the pandemic.

Furthermore, many TCEs across all the visible minority backgrounds in this study also illustrated that being connected with their spirituality and being involved in religious practices was a source of relief for them. Before COVID-19, Ammar announced: “I wouldn’t say it [coping] was easy, but God helps”. This ideology was prevalent amongst participants after COVID-19 as well. Anna reported: “The spiritual part also helps a lot. It helps in that part, that mental strength so that you can be strong when you must face ‘X’ situation”. L.A additionally established: “Praying [helps], and you know finding solace in the promises that […] I have with God”. Therefore, praying and remaining close to their respective faith became a coping mechanism to mitigate stressful situations of caregiving and employment prior to and after COVID-19.

Participants from this research also brought to light that having a strong support system from family and friends aids in managing stress levels. Martina, a TCE of South American ethnicity stated: “My mother has helped me a lot, because right, she’s my mom but she’s also my friend”. Likewise, L.A voiced: “We paid a lot of attention to the family unit […] we spend a lot of time now on Zoom and [chuckles] WhatsApp calls with family and friends”. Fernanda, another participant of South American ethnicity, also expressed: “I just connect with my friends […] going with my family or playing with my nieces that kind of helps me […] step outside of that kind of um stressors”. Hence, many visible minority respondents found social support from their family and friends to better manage their well-being while engaging in transnational care and paid employment.

The final commonality in relation to coping strategies amongst the participants in this study was exercising and meditation both before and after COVID-19. Interviewees shed light on how staying active and instilling a routine exercise plan is a crucial factor to combat the pressures associated with personal and professional lives. For example, in an interview that occurred prior to the pandemic, Lulu voiced how working out has been a coping strategy for her. She stated: “I enjoy going to the gym. I [would] live at the gym if I could”. Nogi also mentioned: “I do meditation […] which is good”. Correspondingly, after COVID-19, there were analogous patterns noted as L.A spoke: “I actually started doing a lot more exercises [...] that helped”. Similarly, Alexandra, a TCE of South American heritage further supported this notion and expressed: “Having an exercise routine [helps] […] sometimes when I’m having a very emotional time and I’m feeling down, to put it that way, I try meditating a lot”. Therefore, the data of this research depict that exercising and meditation is an important tactic that TCEs implement in their lives to support both their mental and physical health.

## 4. Discussion

The findings of this research indicated deskilling of the employed immigrants that are living in London and providing transnational care. It is pivotal to note that the inclusion criteria of this study included being employed or “volunteering”, as the lack of Canadian human capital or work experience and language barriers are among the largest barriers to gaining employment in Canada [45,46,47]. An article by Wilson-Forsberg and Sethi [45] highlighted that newcomers to Canada volunteer to gain experience working in a Canadian setting and to have a Canadian professional status. The study also disclosed that while volunteering can be helpful for immigrants in fostering social connections, it does not significantly improve their economic integration [45]. Ward [46] additionally explains that many skilled immigrants in Canada experience unemployment, underemployment, and find themselves doing jobs that are not commensurate with their educational background. In 2023, the unemployment rate for Canadian-born individuals was 5.0%, whereas for recent immigrants it was 8.2% [47]. The unemployment numbers have always been at the highest disparity for immigrants who recently landed in Canada [47]. Subsequently, even after living in Canada for more than a decade, immigrants that hold a university degree are more likely than Canadian-born individuals to be working in low-skilled occupations [48]. Intersectionality analysis of the visible minority participants with Pakistani, Syrian, African, and South American backgrounds living in London reveals that they were oppressed due to the simultaneous intersection of their race, ethnicity, and immigration status. Systems of oppression like racism and non-recognition of work experience of the TCEs from their country of origin contributed to their oppression and impacted their ability to provide transnational care.

### 4.1. Feelings Associated with Transnational Care

This research illustrated that amidst the financial challenges associated with transnational care, most participants from all four visible minority groups in this study still reported feeling content and a sense of reward for being able to provide caregiving to their family and friends. Although not specific to transnational care, the findings of this study are consistent with the results of research by Henriksson et al. [49] that aimed to assess the feelings of family caregivers undergoing palliative care. Henriksson et al. [49] reported that family caregivers experience a major source of reward for being helpful to their care-recipients and that they also felt satisfied, content, and proud for being able to handle their caregiving duties despite their own unique burdens. A systematic mixed methods review by Bei et al. [50] supports this research and reveals that transnational caregivers observed feelings of personal satisfaction, pleasure, and happiness once they had provided care to their relative with a long-term illness, disability, or frailty. Other scholarly work by Weisser et al. [51], Hochwald et al. [52], and Grant et al. [53] also explained that caregivers perceive the duty of care to play a significant role in their life and feel a sense of satisfaction and reward after providing caregiving as they tend to be appreciated from other members of their family.

Another commonality that was noted among many of the Pakistani, Syrian, African, and South American descent TCEs in this study was that they wished they could provide more caregiving. Participants explained that they hope to engage in more financial caregiving in the future but there were feelings of regret as TCEs disclosed that they felt they could be doing more to give back to their relatives and friends in their home nation. This is similar to the results of an article by Baldock [54] that explored the feelings of immigrants providing transnational care to their elderly parents who remained in their country of birth. Baldock [54] found that while participants engaged in caregiving by calling their care-recipients and doing return visits, they still experienced feelings of regret for not being able to do more. Allard and Whitefield [55] subsequently reveal that caregivers feel drained because of their employment responsibilities and experience feeling guilt for being unable to provide unpaid care to their parents or other relatives. The paper also reports that family caregivers have high expectations of themselves and when they fail to meet their own standards due to work pressures and other constraints, feelings of regret arise for not being able to do more [55].

Moreover, there were some differences observed regarding the feelings of TCEs associated with transnational care as a few interviewees in this study from Syrian and African origin voiced that caregiving can be emotionally draining and overwhelming. This is consistent with a paper by Lyeo and Williams [56] that aimed to assess the experiences of nine Korean Canadian employed caregivers in the Greater Toronto and Hamilton region in Canada. Thematic analysis of the research suggested that all nine respondents faced a form of emotional distress [56]. Specifically, carers mentioned that they felt feelings of burnout, exhaustion, and fatigue because of their workplace and caregiving duties [56]. Other existing literature by Bauer and Sousa-Poza [57] and Schulz et al. [58] also demonstrated that caregivers tend to experience distress and frustration, which is attributed to the illness of the care recipients, employment, and caregiving demands.

### 4.2. Employment Experiences of TCEs

The findings of this research also determined that visible minority TCEs work in occupations that are not desirable to them out of the necessity of providing care and meeting their needs in Canada. Many participants discussed that the language barrier significantly impacts their employment and due to limited proficiency, they end up accepting a job that is low-paying or even demeaning for them. Existing research by Farrell et al. [59] and Corbel et al. [60] exhibited that oral and written communication skills are crucial not just for high-paying jobs but also for middle- and low-income jobs. Like the results of the current research, Carlsson et al. [61] publicized that many immigrants face hurdles in securing their desirable jobs as they are likely to have limited language proficiency in the host country. A study by Batalova et al. [62] backs up these ideas and explains that immigrants with limited skills in English are two times more likely to work at an unskilled workplace in comparison to their English-proficient counterparts.

In addition, the current study sheds light on how many visible minority TCEs of Pakistani, Syrian, African, and South American descent described that there were fewer working opportunities and decreased income because of COVID-19, which hindered their ability to do more financial caregiving. Comparatively, research by Crayne [63] and Bartik et al. [64] explained that the onset of COVID-19–generated lockdowns in 2020 led to hundreds of thousands of businesses being massively disrupted. Wu et al. [65] revealed that the dual responsibilities of being an unpaid carer and an employee made individuals more vulnerable to facing the hardships associated with mass layoffs and business closures. According to Statistics Canada, 12.5% of the employees in Canada were laid off in 2020 [66]. Moreover, a report by the International Alliance of Carer Organizations determined that 56% of caregivers in Canada experienced negative financial health impacts as the pandemic impacted their employment status and increased demands for caregiving [67]. Although not exclusive to transnational caregiving, research by Beach et al. [68] was consistent with the patterns observed in the current study as they reported that the financial well-being of family caregivers after the pandemic was worse and that they were more likely to worry about their financial health as COVID-19 affected their income and employment situation.

Furthermore, while most of the visible minority TCEs in this research reported that there were fewer work opportunities and less income during the pandemic, there were a few participants of African descent who reported the opposite and revealed that they observed an increase in work demand after COVID-19. Nevertheless, it is crucial to note that some of these respondents were working in essential services after the pandemic. This is comparable to the findings of Bell et al. [69] who investigated the experiences of essential workers during the COVID-19 lockdown in New Zealand. The results of the cross-sectional survey revealed that essential workers had a greater workload relative to those employed in non-essential occupations [69]. Likewise, Brophy et al. [70] undertook qualitative research to document the lived experiences of essential workers post pandemic. The results of the study indicated that participants faced increased workloads as they were working longer hours to keep up with their paid duties and mitigate the shortages in staff [70]. Correspondingly, a report entitled “People at Work 2022: A Global Workforce View” published by the ADP Research Institute disclosed stark differences in the experiences of remote workers compared to on-site workers during the pandemic [71]. The report illustrated that home workers were putting in three more hours for free every week than those who had to go to their place of employment in person [71].

### 4.3. Coping Strategies for Well-Being

Respondents from all the visible minority groups in this study voiced that one of the main coping strategies that they implemented in their lives to improve their well-being both before and after the pandemic was keeping busy. Visible minority TCEs of Pakistani, Syrian, African, and South American origin specifically mentioned that they distracted themselves with work and tried to keep their mind occupied to prevent negative thoughts and feelings. A study by Williams et al. [72] also emphasized this when they carried out five online focus groups to explore the experiences of the COVID-19 pandemic in the United Kingdom. Williams et al. [72] reported that a main coping tactic that respondents used to avoid a low mood was staying busy with education, work, and other social activities. Another research by Ogueji et al. [73] declared that being occupied with employment and academic life was a major coping mechanism for many individuals during the pandemic because it distracted them from feeling negative emotions and kept their concentration on their responsibilities. It is important to point out that while staying busy and productive may result in feeling empowered and confident, there can be detrimental impacts on the mental health of individuals if they are suppressing and avoiding their unwanted feelings [74]. Specifically, staying busy and productive releases endorphins, or happy hormones in humans; nonetheless, if individuals purposely internalize their emotions for a long period of time, it may manifest into frustration, anger, and other negative mental health states [74].

The interviewees in the current study also elucidated that they mitigated the stress from their dual responsibilities of care and work by staying in touch with their religious practices and spirituality. Existing scholarly research by Xavier and Esperandio [75], Britt et al. [76], and Pearce et al. [77] determined that religious and spiritual coping is a pivotal coping strategy for family caregivers and is linked with greater carer satisfaction and less carer burden. Studies distinguish that praying and reading religious texts can improve the mental health of those providing emotional, financial, and personal care to their family members with chronic illnesses [75,76,77]. Previous literature by Marks and Dollahite [78] also explained that religiosity and spirituality promote the psychological well-being and subjective health of caregivers. Further, Sen et al. [79] looked at the role of religion on the mental health of caregivers during the COVID-19 pandemic. The findings of Sen et al. [79] strengthen the results of this paper as they publicized that beliefs connected with religion and spirituality are a promotive factor that correlated with improved mental health and are a positive coping strategy during times of social disruption such as COVID-19.

Furthermore, the visible minority interviewees in this research highlighted that staying in contact with their family and friends helped them better manage their well-being. The TCEs from all four ethnicities expressed that talking with their loved ones on video and phones allowed them to feel comfort and ease. This is analogous to the research by Yang et al. [80] that examined the role of coping tactics in the promotion of psychological well-being during the pandemic. The results of the study suggested that family support is positively associated with better mental states [80]. Subsequently, an interpretative phenomenological analysis was conducted by Xiuxiang et al. [81] to assess the coping strategies of 14 family caregivers in China. The results of the study acknowledged that family support from other relatives facilitated them in emotionally coping with their caregiving responsibilities. Mariani et al. [82] strengthened these perspectives and illuminated that family support decreases a sense of isolation and plays an exclusive role in combatting negative mood symptomatology during international health crises.

The last commonality among the TCEs of Pakistani, Syrian, African, and South American descent in this paper regarding coping mechanisms was staying active. Most of the visible minority TCEs from all backgrounds affirmed meditation and exercise was important tactic that inhibited stressful thoughts associated with their employment and caregiving services. Equivalently, a descriptive study by Hamed et al. [83] reported that physical exercise is a good coping mechanism that helps decrease stress, anger, and frustration. Although not exclusive to transnational care, research by Dyrbye et al. [84], Singh and Afroz [85], and Jadhav [86] illustrates that exercise alleviates burnout and that individuals who stay physically active have low levels of depression, anxiety, and stress levels in comparison to those with sedentary lifestyles.

### 4.4. Limitations

The scope of this research contains some limitations. As with most qualitative studies, this study may not be generalizable to Pakistani, Syrian, African, and South American origin TCEs beyond the respondents that were recruited for this research. This is also because the experiences of TCEs in this study are specific to a particular time in an urban area like London. Hence, the results may not be representative of TCEs in other sub-urban or rural settings elsewhere. In addition, it should be noted that most participants disclosed caring for their parents with chronic health conditions such as diabetes, and other immediate and extended family members. However, there was a lack of understanding of the experiences of TCEs who care for loved ones with a physical disability; thus, the finding of this research may not be illustrative of the circumstances of other visible minority TCEs. This research used qualitative measures to scrutinize the tacit lived experiences of TCEs from various ethnicities. Future studies may use quantitative methods to obtain a more precise understanding of transnational caregiving in the times after the pandemic. Future directions should also be geared towards understanding the impact of deskilling on TCEs and their ability to send money back home as well as caring for their family in the country of relocation.

## 5. Conclusions

As a result of immigration, the Canadian labor force grows yearly as immigrants pay taxes and spend on goods, housing, and transportation [10,87]. Immigrants also offset the trends of the rising retirees in the Canadian society, drive innovation, and attract investments in Canada as they make up for half of the nation’s Science, Technology, Engineering, and Mathematics (STEM) degrees [10,87]. Despite the essential role that immigrants play in the well-being of the Canadian economy, workplaces lack inclusive and carer-friendly workplace practices and policies, especially for visible minority TCEs [3,6]. This research brings forth evidence that, among Pakistani, Syrian, African, and South American origin TCEs living in London, Ontario, most of the participants experience a sense of reward and satisfaction for being able to help their families overseas. Nonetheless, participants expressed their wish to be able to do more transnational care for their care-recipients as financial limitations currently hinder their caregiving capabilities. Even though most of the participants experienced a commonality of feeling happy after fulfilling their caregiving obligations, there were some interviewees of Syrian and African origin who reported feeling frustrated and overwhelmed with their dual duties of informal transnational care and paid employment. In terms of the employment experiences of TCEs before and after the pandemic, respondents stated that they took up any work in the labor force that they were offered as they had to help their relatives back home and that due to the language barrier, they worked at occupations that are viewed to be demeaning for them from a cultural lens. Many participants further highlighted that they experienced a lack of work opportunities and a decreased income post pandemic. In contrast, a few interviewees of African descent expressed that the workload increased for them after COVID-19. However, it is crucial to know that many of these interviewees who reported a rise in workload were working in essential services during the COVID-19 crisis. Finally, TCEs across all four cohorts reported that keeping busy with work, praying, family support, as well as staying active were coping strategies that they implemented in their lives both before and after COVID-19 to sustain their mental and physical well-being. The current study reveals remarkable similarities across the recruited visible minority TCEs and limited differences. It is vital to recognize that differences among these Pakistani, Syrian, African, and South American origin TCEs living in London may have been masked by the common challenges of making ends meet while striving to support loved ones elsewhere during the times of COVID-19. Moreover, due to the lack of accommodations for caregivers in the workplaces, skilled workers end up leaving their employment to manage their caregiving role [5]. The consequence of this is substantial for Canada’s economic stability as CAD 1.3 billion are lost in productivity every year [5]. This research advocates for the implementation of gender-sensitive policies and CFWPs like the CSA and ISO Standards in workplaces so that there can be accommodations for immigrants that provide transnational caregiving [5]. These Standards will allow TCEs to effectively contribute to the Canadian economy while being in a stable financial, emotional, and physical state. Implementing these Standards will lead to much-needed resources for TCEs such as financial assistance, caregiving responsibilities leave, flexible working hours, non-contiguous paid leave, consolation services, as well as communication and networking opportunities [5]. This study urges decision-makers to recognize the need for CFWPs so that TCEs can be productive, supported, and most importantly, healthy [5].

## Figures and Tables

**Table 1 ijerph-20-06800-t001:** Participant characteristics.

Participant Characteristics	Number of Participants
**Sex**	
Male	15
Female	14
Care-recipient types	
Immediate family (parents, children, in-laws, spouse) only	10
Extended family (cousins, nephews, nieces, spouse’s family) only	4
Both immediate and extended families	15

**Table 2 ijerph-20-06800-t002:** Origin of the participants.

Origin		Number of Participants
South America	Barbados	2
	Venezuela	1
	Haiti	1
	Colombia	10
Africa	Kenya	1
	Nigeria	6
	Uganda	1
	Zimbabwe	1
Syria		4

## Data Availability

The data and materials that support the findings of this research are made available from the corresponding author upon reasonable request.

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
