# Peer review of "Commonalities and Differences in the Experiences of Visible Minority Transnational Carer–Employees: A Qualitative Study"

_ijerph, 2023, doi:10.3390/ijerph20186800_

Round 1

Reviewer 1 Report

Dear authors

I would like to thank you for giving me the opportunity to review the manuscript entitled “Commonalities & Differences in the Experiences of Visible Minority Transnational Carer- Employees: A Qualitative Study”. This study is a secondary analysis of qualitative data and explored the commonalities and differences among the experiences of visible minority TCEs before and after COVID-19. This study is well written and can be a basis for further study design for the health of this high-risk group. I have some comments as follows:

Abstract

- You should briefly mention why this study was worth doing.

- Please write the full form of abbreviations in the first use.

- How was the sampling method?

Methods

- Have other traditional data collection methods been used in ethnography such as observation? If not, give a convincing reason for it.

- I think that your study should be a secondary analysis on qualitative data.

- Researcher characteristics and reflexivity - Researchers’ characteristics that may influence the research, including personal attributes, qualifications/experience, relationship with participants, assumptions, and/or presuppositions; potential or actual interaction between researchers’ characteristics and the research questions, approach, methods, results, and/or transferability

- Techniques to enhance trustworthiness - Techniques to enhance trustworthiness and credibility of data analysis (e.g., member checking, audit trail, triangulation); rationale

Reviewer 2 Report

I enjoyed reading the paper.  It is well written with few typographical errors. The methodology is clear and well-described.  This is interesting and helpful research.

However I lost some understanding of the overall focus of the paper.  It would be helpful to define what you mean exactly by TCE, and also to spell the abbreviations out in full.  The introduction focuses on carer employees in Canada, but it is not obvious from the findings as presented, or the participant responses, that they are caring for disabled / older etc family members.  The findings seem to link to the caring duties of any person who is a migrant sending money etc back to the home country.  It would be helpful to link the introduction to the findings and then take that through into the discussion.  As I read the article I was confused and could not link the identity of transnationals carrying out 'care-giving duties', beyond child care etc, to care-giving duties related to disabled / health impaired or older family members.  The definition of TCEs needs to be clearer, so that the reader can relate the introduction and the proposed focus of the article into the findings and discussion.

For example, the feelings associated with transnational care in the findings need to be related to care-giving and its impact - and not just the impact that a migrant worker may experience from not seeing children or family members.  How are these feelings different as care-givers to disabled / health impaired people than to people caring for non-disabled family members.  This distinction would really add value to the paper, and would allow the golden thread, introduced in the introduction to follow through the paper.

Please check out first few lines of discussion section - 480 - 484.  Is this supposed to be written in this way?

Also the conclusion focuses on the TCE and employment opportunities and adaptations for carer-friendly work.  It may be helpful to try to focus the findings and link in discussion more about this, to give a holistic feel to the paper.

I look forward to receiving these revisions.

Round 2

Reviewer 1 Report

Dear authors 

Thank you for addressing my comments.